# Balloon drift estimation and improved position estimates for radiosondes

Ulrich Voggenberger[1], Leopold Haimberger[1], Federico Ambrogi[1], Paul Poli[2]

[1]Department of Meteorology and Geophysics, University of Vienna, Vienna, 1090, Austria

[2]European Centre for Medium-Range Weather Forecasts, Bonn, Germany

*Correspondence to*: Ulrich Voggenberger (ulrich.voggenberger@univie.ac.at)

**Abstract.**

When comparing model output with historical radiosonde observations, it is usually assumed that the radiosonde has risen exactly above its starting point and has not been displaced by the wind. This has changed only relatively recently with the availability of Global Navigation Satellite System (GNSS) receivers aboard the radiosondes in the late-1990s, but even then the balloon trajectory data were often not transmitted, although this information was the basis for estimating the wind in the first place. Depending on the conditions and time of year, radiosondes can sometimes drift a few hundred kilometres, particularly in the mid-latitudes during the winter months. The position errors can lead to non-negligible representation errors when the corresponding observations are assimilated.

This paper presents a methodology to compute changes in the balloon position during its vertical ascent, using only limited information, such as the vertical profile of wind contained in the historical observation reports. The sensitivity of the method to various parameters is investigated, such as the vertical resolution of the input data, the assumption about vertical ascent speed of the balloon, and the departure of the surface of the Earth from a sphere. The paper considers modern GNSS sonde data reports for validation, for which the full trajectory of the balloon is available, alongside the reported wind. Evaluation is also conducted by comparison with ERA5 and by conducting low-resolution data assimilation experiments. Overall, the results indicate that the trajectory of the radiosonde can be accurately reconstructed from original data of varying vertical resolution and that the more accurate balloon position reduces representation errors, and, in some cases, also systematic errors.

## 1 Introduction

Prior to the availability of remote sensing techniques, upper-air measurements of air motions were widely collected using Lagrangian perspectives, with weather balloons (e.g., Dutton, 1986). The uncertainty of such upper-air observations depends not only on the measurements themselves but also on the availability and quality of associated metadata and measurement position: this is generally associated with so-called representation errors (e.g., Kitchen, 1989). As weather balloons drift with the wind during their travel, including ascent, they can thus be displaced over large distances (**Figure 1**), in some cases more than 400 km from their launch base (e.g., Seidel et al., 2011). Precise knowledge of the balloon position is particularly important in regions of sharp horizontal gradients, e.g. near mountain ranges or near jet streams. Tschannett (2003) and Steinacker et al. (2005) noted that apparent superadiabatic vertical lapse rates in Foehn events disappeared after the balloon displacement had been taken into account. For operational monitoring, detailed information regarding the balloon trajectory was generally not recorded or not transferred via the data distribution networks until the advent of Global Navigation Satellite Systems (GNSS). Even later, when GNSS sensors became available, the information collected was often not transmitted, although the wind data was calculated directly from it (WMO, 2021), as there was no available space in the alphanumeric codes. This became possible with the (ongoing) migration from alphanumeric codes to Binary Universal Form for the Representation of meteorological data (BUFR), allowing also the reporting of many more levels in the vertical (Ingleby et al., 2016). Only since the 2000s efforts have been made to take into account the balloon drift in modern observation processing of GNSS sondes, with beneficial results (e.g., Keyser, 2000; Laroche and Sarrazin, 2013; Ingleby et al., 2018).

Radiosonde measurements are used in a variety of applications, including near-real-time by forecasters and Numerical Weather Prediction (NWP), but also for air pollution or other scientific investigations, including climate monitoring (e.g., Dabberdt and Turtiainen, 2015). The production of climate reanalyses that directly assimilate radiosonde observations, such as ERA5 (Hersbach et al. 2020), is expected to benefit from more accurate historical balloon position data, similarly to NWP. In this regard, the location precision of the assimilated measurements should be commensurate with the horizontal resolution of next-generation reanalyses (~10 to 20 km globally, e.g., Hersbach et al., 2022). At such resolutions, assuming vertical ascents for a balloon that is displaced by a couple of hundred kms would amount to comparing the balloon measurements with model values that are 10 or more grid boxes away, which is clearly suboptimal. Resolving this situation requires, for historical soundings, to reconstruct the balloon trajectories from the little information that is available (Stohl, 1998). In many cases, this information only consists in the vertical profile of wind, as discussed later in the paper.

Section 2 describes the data and a method to calculate the balloon drift from historical radiosonde ascent data. Details of the technical implementation, with python code and test data, are provided in section 3. Section 4 presents validation results, including several sensitivity analyses to explore the robustness and accuracy of the approach. Sections 5 and 6 show

evaluation results, using two different approaches, whereby the beneficial impact of the more accurate balloon position is
demonstrated. Section 7 includes a discussion and conclusions.
**2 Data and methodology**

**2.1 Radiosonde data**
Radiosonde data used in this work are obtained from the Integrated Global Radiosonde Archive (IGRA), Version 2 (Durre et
al., 2016) and via the Copernicus Climate Change Service (C3S) Climate Data Store (CDS). High-resolution radiosonde data
used for validation are obtained in BUFR format from the National Centers for Environmental Information Radiosonde
Archive (NOAA NCEI).

The quality of the available wind data depends on their encoding and the method used to track the balloons. Measuring
techniques for upper-air winds have changed significantly over time, with a clear general trend towards improvements in
quality, thanks to removal of procedural errors, in particular (e.g., Crutcher, 1979), noting also improvements in the accuracy
of encoding, with evolution of the data formats. All these changes are described in the WMO Publication Nr. 8, Guide to
Meteorological Instruments and Methods of Observation, published since 1954 by the WMO Commission for Instruments
and Methods of Observation (CIMO; WMO, 2021). Regarding changes in the measurements of wind and balloon positions,
there are three important distinctions to be made.

The first distinction concerns the sensing apparatus: non-GNSS versus GNSS sondes. Early observations used only ground-
based tracking, e.g., by theodolite, which was fairly accurate but could lose the balloon early during cloudy or high wind
speed conditions, and relied on an assumed ascent rate if, like in most cases, a single theodolite was used (e.g., Favà et al.,
2021). From the mid-1950s onward, radar tracking or radio-positioning of the radiosonde became standard. Wind
components were then calculated from the measured position and time differences.
In the 1990s, GNSS modules were introduced to track the horizontal and vertical position of the sensor at high frequency,
thanks to improvements and miniaturisation of the electronics. The resulting data were then used to calculate the wind
variables, but the position data were not transmitted to the global network and are therefore not available in global databases
in                 most                 cases                 until                 2014.
The higher frequency of observations exchanged in recent years can expose the pendulum motion of the sonde beneath the
balloon in its observed position (Ingleby et al, 2022). In our experimental cases, we did not observe any significant effect of
the pendulum motion, its magnitude being generally much smaller than the wind advection displacements, suggesting it does
not appear to need to be taken into account to first order.

The second aspect is the determination of altitude. Prior to GNSS observations, altitude was determined by three different
methods: ascent speed estimation, pressure sensors, and vertical radar or radio-positioning, with continued efforts to increase
the quality of observations over time. Ascent speed can be affected by many factors, and Murillo et al. (2005) estimated a
scatter in linear ascent rates of about 5% about the mean value for pilot balloons, after using double theodolites to conduct
measurements to measure the balloon height during ascent.

The third aspect is the data format used for transmission. Essentially two main message systems have been used to transmit
the observed radiosonde data: Traditional Alphanumeric Code (TAC) and BUFR. The main difference is that BUFR allows
not only for a much higher vertical resolution (up to 1 second frequency, corresponding to approximately 5 m altitude
difference) but also for a higher coding precision. The BUFR messages report wind direction with a resolution of 1-degree,
whereas TAC messages report wind direction to the nearest 5-degrees. In addition, time and three-dimensional position
information is only transmitted via BUFR but not via TAC. TAC messages typically also include data only on mandatory
and significant levels. Mandatory levels are a set of predefined pressure levels. Significant levels for wind are added as
needed before transmission so that the wind speed does not deviate by more than 5 m/s from linearly-interpolated values,
according to the above-cited WMO CIMO guide.
There are also thermodynamically significant levels, which refer to specific levels of atmospheric pressure at which
significant changes in temperature, humidity or other thermodynamic properties occur. Most transmitted radiosonde profiles
include some of these.
**2.2 Quality control**
The following steps are taken to exclude outliers:
● For wind speed, we applied a range check, with wind speed limited to 150 m/s, a value that is rarely reached, even
in strong upper-level jets.
● For temperature, needed for geopotential calculations, we relied on the IGRA2 quality control (Durre et al. 2018)
that already removes gross errors. A range check was also applied, with temperature limited to between 173 and
373 K, to verify that the data were read correctly and avoid possible encoding errors in the messages.
Observations that fall outside these limits are not processed further, to avoid degrading the quality of the output (balloon
trajectory).
It was investigated whether additional quality control measures would improve performance and the validation of the RMSE
differences discussed in section 5. To improve outlier removal, we filtered the observations based on the 1st and 99th
percentiles of the differences observations minus ERA5 forecast (these differences are called background departures
afterwards). This was completed in two stages: once for each level, and then again for the entire set of available wind speed
and temperature data. However, neither of the two versions improved the RMSE differences. Rather, we found that the
background departures were often large enough to be discarded just in the interesting cases of strong but plausible
displacements. The reason was not always the displacements themselves but also the fact that large lateral displacements can
lead to large height errors in profiles from non-GNSS Russian radiosondes, since those have no pressure sensor but rely on
radar heights (Kats et al. , 2005). However even for these sondes, we found that taking into account the balloon drift reduces
the differences to the ERA5 background forecasts.
The results presented in section 6 include the standard quality controls applied during data assimilation experiments, as
detailed in the technical documentation published by ECMWF (2023).

Filtering radiosonde data before the displacement calculation based on the number of available observations per profile is
recommended. A profile should not be too coarse and should not start too high above the ground. For the experiments
conducted in this study, the limit for the initial observation was set at 1500 m above the release station height.
**2.3 Estimation of the balloon trajectory**
The balloon position is calculated relative to the launch position (so-called base coordinates), as latitude displacement and
longitude displacement (decimal degrees). For each vertical level, these two values can be added to the base coordinates to
obtain the new (latitude, longitude) position at the given level. The same approach applies to the reconstruction of the
measurement times at all levels. This practice conforms to the BUFR encoding standard.

For the position calculation, the same simple physical laws that have been used to derive the reported wind components are
applied. Only a few initial parameters are necessary for this:

● station coordinates or starting point of the sonde, (latitude and longitude) ;
● wind vector (zonal and meridional components, noted respectively u and v), measured by the sonde at different
pressure levels;
● measurement time (t) at different pressure levels.

These variables enable calculation of how long the sonde was exposed to horizontal wind, and therefore can be used to
estimate the displacement of the sonde.
Especially older datasets often only contain the starting time of the ascent, time information is not available for any of the
reported pressure levels.
To estimate the time elapsed since the release of the balloon, three variables are needed:

● the reported pressure levels (generally available from radiosondes) or heights (generally available from so-called
PILOT balloons, also called PIBAL),

14                                                                5

● the sonde ascent speed.
● the surface pressure or station height (not strictly needed for displacement calculation since first level is typically
157       reported quite close to the surface)


PILOT or PIBAL profiles provide an estimate of height at each level, from which the time at each level can be reconstructed,
assuming a given ascent speed. However, for multivariate soundings (radiosondes reporting temperature and wind), observed
pressure is often the only information available regarding the radiosonde vertical position. In such a case, the pressure profile
needs to be transformed to a height profile. This can be done assuming a piecewise constant temperature gradient between
the levels in the profile. The calculation of the vertical gradient of temperature with respect to altitude from the vertical
gradient of temperature with respect to pressure is shown below in **Formulae 1** and **2**. Subsequently, **Formula 3** indicates
how this information is used to determine the heights of all pressure levels. If the height information is already available (e.g.
PILOT data), those steps can be skipped.

The vertical resolution of the available data varies. While early ascents often contain even less than the mandatory levels (16
levels), recent data in high resolution BUFR are available on 3000 levels or more. The sensitivity of displacement
calculations       to       vertical       resolution       is       investigated       later       in       this       paper.
If a single mandatory level is missing within the ascent range, then the displacements are not calculated; we consider that too
much information is missing in such a case. If a level was not mandatory in historical data (e.g. 70 hPa, 250 hPa, 925 hPa),
this rule does not apply to the data. However, an early termination of the vertical ascent is not an issue, then the
displacements are only calculated up to the highest available level.

The determination of the sonde's ascent speed is more uncertain. It depends on some variables that are poorly determined or
unknown, such as the air vertical wind speed and the weight to buoyancy ratio of the probe and the balloon. Deviations in the
filling level of the balloon, the air resistance of the balloon skin, as well as the ambient temperature and the balloon gas
temperature further influence the ascent speed. A review of some of these factors was made by Favà et al. (2021).

Using data from recent sondes, our study of the data with known altitude time series indicates that the rate of ascent varies
mostly between 2 and 10 m/s. Within this large range, **Figure 2** shows that the mode of the distribution of ascent speeds is
around 5 m/s. **Table 1** further indicates that the interquartile range is 2 m/s (i.e., from 4 m/s to 6 m/s). These findings are
consistent with other sources (e.g., Seidel et al., 2011). These statistics represent global fluctuations in the ascent speed of
weather balloons.

Over short time scales, **Figure 3** indicates the vertical velocity of the probe fluctuates substantially. This is true both within a
single ascent and also between different ascents. Near the ground and above the tropopause the fluctuations are largest.

17                                                                6


Given the considerations above for historical balloons, one must recognize that the vertical speed can only be estimated in
most cases, and will always lead to significant deviations as compared to measurements obtained from high-resolution data.
Note the high vertical resolution shown in **Figure 3** is hardly reached in ascents before the year 2000. This also means that if
only mandatory levels are available, the fluctuations in average ascent speed at each available level are smaller, due to the
longer averaging intervals.

**Figure 2** and **Figure 3** show that an assumed ascent rate of 5 m/s agrees well with the observed mean value. To counteract
the effects of this fluctuating parameter, an attempt was made to use a height-dependent function instead of a constant speed,
which represents the annual average over more than 100 stations.

As part of this experiment, a polynomial model was also tried, in an attempt to improve the accuracy of the average ascent
speed. The resulting displacements showed, however, very little improvement (i.e. smaller differences to GNSS measured
displacements), indicating that the assumed vertically constant ascent rate of 5 m/s is a sufficient approximation.

As a next step, it is necessary to calculate the height profile from temperature and pressure information. For this step, we use
the formula for a dry atmosphere with piecewise constant lapse rate (Alexander and de la Torre, 2011). Relative humidity
could also be considered by using the virtual temperature, but it is often not available for early ascents and we also found
that the differences in resulting displacements were small. For the first level, the International Civil Aviation Organization
(ICAO) standard atmosphere lapse rate of -0.0065 K/m is used. For all subsequent steps, the temperature gradient is
calculated directly from the temperature and pressure profile (mean values for each layer "i").

The height profile is then used to calculate the time interval spent by the sonde between the noted levels. It can be estimated
using the estimated vertical velocity mentioned earlier.
These time intervals are then used to determine the transport of the balloon according to the mean wind inside the layer
between the levels i to i +1, see **Formula 4**.

Afterwards, this distance is converted into latitude and longitude using either the inverse Haversine method on an assumed
sphere, or the forward transport function on the "WGS84" ellipsoid. The difference between the two transport functions is
found to be practically invisible for smaller observed displacements (see **Figure 4**). Nevertheless, the ellipsoid option is used
as it should deliver higher accuracy results. Finally, the resulting latitudes and longitudes are subtracted from the base
coordinates to obtain the displacements.

Particular care is required when using reported wind direction near the North or South Pole. For example, when crossing the
North Pole, a radiosonde in a southerly airflow (prior to the crossing) finds itself in a northerly airflow (afterwards). So far,
only TAC has been used at the South Pole station, which means that the wind components are reported according to the
launch position, not to the actual position, and is thus constant during the ascent. We calculate the displacements in x and y
direction valid at this position and then convert that back to lat/lon positions and displacements.
The WMO Manual on Codes states that for stations within 1° of either pole wind direction shall be reported in such a way
that the azimuth ring shall be aligned with its zero coinciding with the Greenwich 0° meridian. There is currently an attempt
to update this advice for BUFR reports, such that wind direction should be reported relative to the current reported longitude
- to help in NWP use of such winds. Before comparing winds from the South Pole station with NWP fields they should have
their direction adjusted when the drift positions are calculated, but note this was not done in the present work.
Although the principle of displacement calculation is similar to the method presented in earlier work on this topic (Laroche
and Sarrazin, 2013), we use different input data for height information. Instead of using the average ascent time for each
standard level, we calculate the times for each available level using the mean lapse rate for the representative layer.
Aberson (2017) applied a similar approach for dropsondes, albeit with a different way of calculating the vertical velocity.
Both of these methods are successful and promising, and for the purpose of this method they have been used as the basis for
reconstructing          the          trajectories          as          best          as          possible.
**3 Implementation and availability**
The software necessary for the creation of calculated balloon trajectories can be found in the Python package rs-drift:
● https://zenodo.org/records/10663306
● https://pypi.org/project/rs-drift/
Examples on how to use it are available in all repositories as an IPython notebook "rs_drift_example.ipynb".

In addition to the coordinates of the launch site or station in degrees latitude and longitude, the trajectory function requires
profiles of four input variables in the right units: temperature [K], pressure [Pa], zonal wind (u) [m/s], meridional wind (v)
[m/s]. It accepts only input which is sorted in ascending order.
```
trajectory = rs_drift.drift.trajectory(lat,lon,temperature,u,v,pressure)
```

The function returns the following output:
```
trajectory == [latitude_displacement, longitude_displacement, seconds_since_start]
```

All those output variables are numpy arrays, with one element for each pressure level - with the same length as the input
data. For PIBAL ascents, the geopotential height must be provided as an additional keyword parameter.

It is possible to experiment with input data. If humidity information is available, the virtual temperature can be used instead
of the observed air temperature. Also if more information of the balloon's mean ascent rate is present, this should be used as
input in the additional arguments. Any approach including proper quality control of input data that is available should be
used to create the best possible estimation of the balloon drift.
The drift of the balloon and sonde compounds is introduced as "displacement" from the starting point (launch site). For
simplicity, the displacements can be added to the base coordinates to obtain the vertical profile of positions of the balloon.

**4. Validation with GNSS radiosondes**
Validation per se is only possible when a trusted source can provide a good reference. Such is the case for modern sondes
equipped with GNSS receivers, when it comes to the recovery of the balloon trajectories. For pre-GNSS radiosondes, a
similar validation would be possible, if only one had available the information about the balloon trajectory. Unfortunately,
this information is available only in rare cases.

The data from the modern GNSS radiosonde data encoded in the recent high-resolution BUFR files are used to verify the
systematic and random errors of the calculated displacements at different pressure levels. This data set contains second-by-
second records of actual positions of the sonde measured by GNSS in the form of displacements, thus enabling the direct
comparison with the calculated displacements.

**Figure 4** also shows that the displacements obtained from GNSS and the displacements calculated from the wind data agree
quite well. The small deviations likely come from differences between the actual (unknown) and assumed (5 m/s) ascent
rate.

**Figure 5** provides an overview how large the displacements typically are and gives profiles of uncertainty estimates for the
calculated displacements. In the troposphere the RMSE is mostly below 0.02 degrees (2.5 km), in the stratosphere it can be
up to 0.1 degrees (12 km). These numbers amount to uncertainties of about one part in five to ten, of the observed variations
(RMS), in the example shown. Still, this is much better than just ignoring the displacement.

These results were obtained by using as input the high-resolution data. For historical radiosondes, only comparatively low-
resolution information is available (in the form of mandatory plus significant levels).
In **Figure 6** and **Figure 7**, the impact of using only mandatory and significant level information is shown. The difference of
displacements in **Figure 6** is minimal, although the displacement is relatively large.
**Figure 7** shows a case of larger differences in relative terms. The overall zonal displacements are large and the winds vary
strongly with altitude. An issue arises when selecting data points with low representativeness from the ascent, particularly
those that are far from the layer average. This can result in less accurate outcomes compared to using averages from less
detailed data. Figure 7 provides a good example of this issue with the v component of wind at original resolution and
mandatory pressure levels only. The method of calculating the displacements itself uses mean wind speeds within the
considered levels. Thus, if the observations are also means of larger vertical height differences, more or less randomly
observed            peaks            become            a            smaller            source            of            error.
Figures 6 and 7 respectively show the range of accuracy of the calculated trajectories quite well. The final displacements
may differ in quality depending on the quality of the observations, the representativeness of the available levels, and the
vertical resolution. All ascents in the validation examples had displacements, which added value in bringing the observation
closer to the true position. The accuracy may vary based on the aforementioned input variables. However, we did not find
any case where using the displacements would lead to a worse position estimate.
**Figure 8** shows the comparison between the displacements of two different data sets - on high resolution BUFR levels and
on the other hand on mandatory levels only. It can be seen that for this subset of ascents there is still much value in the
displacements for the mandatory levels only version. However, it should be noted that more available levels always lead to
better results and the highest possible number should be used in any case.
Many of the older observational reports contain temperature and wind data on different levels. Only at mandatory levels both
variables are available. In this case, interpolation can be performed for the points in between. When applied to IGRA data,
wind data are interpolated to levels of the temperature observations. This allows the input to be maximised to calculate the
best possible displacements.

**5. Evaluation with ERA5**
To evaluate the impact of taking the displacements into account, we compared the observed values from the radiosondes
with the gridded ERA5 data, in one case assuming a strictly vertical ascent, and in the other case assuming an ascent along
the calculated (slanted) trajectory defined by the displacements. The ERA5 fields at hourly resolution and 1° x 1 ° horizontal
resolution were interpolated linearly horizontally to the observations locations defined in either of the two cases mentioned
earlier                            (vertical                            or                            slanted).
These tests and comparisons used the short term forecast of the ERA5 assimilating model, also referred to as "background".
This choice, instead of using ERA5 analyses, was made  to try to maintain as much independence  as possible with respect to
the observations. This choice should largely avoid possible problems resulting from the fact that the observations are also
assimilated into the ERA5 data, given that many other observations were assimilated alongside radiosondes and also
influenced the analysis state. Experimental comparisons to the ERA5 analyses (in contrast to background forecasts) showed
that the analysis data fits significantly better with the vertical trajectory of observation than with the slanted version. This is
to be expected, since radiosondes were assimilated as vertical profiles in ERA5.

**Figure 9** shows the benefit of comparing the radiosonde observations with the *background forecasts* as slanted profiles
instead of vertical profiles. In low layers (below 700 hPa), the displacements are relatively smaller than at higher levels, and
therefore hardly lead to deviations for temperature. In most cases, there is an improvement at levels located above 750 hPa,
though at some stations the improvement is visible already as soon as the sonde reaches 850 hPa, depending on the wind
speed and topography around the station. Typically, the effect is largest in regions with high upper-level wind speeds.
Taking the displacements into account improves the background departure statistics between measurements and ERA5 not
only for temperature but also wind and relative humidity.

For relative humidity, the improvement is confined to levels located below 250 hPa. Above this level, the relative humidity
is generally very low, making it difficult to detect any meaningful difference with respect to the ERA5 background.
It is also important to note that some stations, where the RMSE of the ascents do not show signals of improvement in
temperature, often still show improvement in humidity or wind (or vice versa).

Considering that radiosonde observations make up a larger part of the total observations for the reanalysis in earlier years,
one might think that especially for these years the displacements are more relevant. The data investigation reveals that
improvements of the departure statistics are not greater for earlier ascents than for more recent ascents. The reason might be
that reanalysis fields before the satellite era are more strongly dependent on radiosondes. At these times few other upper-air
observations were available, and radiosonde data were assimilated assuming vertically straight ascents. However, the density
of the input data and the general quality of the reanalysis increased over the time, while the bias in measurements of the
uppermost levels decreased over time. Therefore, the relative importance of representation uncertainties,  with respect to the
two other sources of uncertainties in the comparison (radiosonde instrumental uncertainties and ERA5 background
uncertainties), is larger for more recent ascents. **Figure 10** shows that considering the displacements is beneficial, although
to a lesser extent, also in the early days, when little upper-air information other than radiosondes was available.

Finally, in **Figure 11** there are the results of a global comparison for the year 2000 - like the previous ones, but calculated for
all the available stations. A positive difference again indicates improvement due to taking  the displacements into account.

To give a better insight, the differences of the RMSE are also plotted on a map for the 150 hPa level in **Figure 12**. Warm
colours show improvement for the respective station by applying the displacements, cold colours show a deterioration.
Improvement clearly predominates for the majority of stations. Deteriorations in quality appear less frequent and of smaller
magnitudes than improvements.

**Figure 13** shows the difference of the ERA5 background eastward wind speed in the 1990s at the station location minus the
same wind speed at the displaced location. The differences are sizable in some regions. For example, the weaker wind speeds
above station locations in China would indicate systematically too high observed wind speeds. This effect is large enough to
explain some of the radiosonde wind minus background wind differences, as pointed out by Tenenbaum et al. (2022). This
stresses again the importance of avoiding position errors in historical radiosonde ascents. Without the adjustments, artificial
trends in wind speed from radiosondes would be introduced in some regions when switching from traditional to GNSS
radiosondes.

**6. Evaluation with data assimilation experiments**
Desroziers et al. (2005) proposed a method to diagnose uncertainty statistics of observations in a data assimilation
framework. As indicated in their work, there are important assumptions associated with the approach. Bias contributions
aside, the overall level of uncertainties may be incorrect if, for example, there is significant correlation between observation
random uncertainties and random uncertainties of the background that is used in the data assimilation. A separation of scales
is indeed required in order to disentangle these two uncertainty components. Given the unique importance of radiosondes to
inform on the state of the stratosphere in a background obtained from data assimilation, such as in a reanalysis (e.g.,
Hersbach et al., 2020), there may be some components of the uncertainties (such as radiation) that are present, and possibly
correlated, in the background and the observations. For these reasons, we do not use Desroziers' diagnostics in order to
assign undisputable uncertainties to the radiosonde uncertainties. Instead, we use these diagnostics in order to detect any
changes in the observation uncertainties, which include instrument and representativity uncertainties, owing to the effect of
balloon drift.

To this end, we run two data assimilation experiments, using a simplified data assimilation setup. Simplifications are
required in order to make such an undertaking numerically affordable. Otherwise, so-called 'full' data assimilation
experiments, using all observations at the maximum resolution, are indeed too costly to conduct, if only for such an
evaluation. The simplified data assimilation setup is based on the ECMWF Integrated Forecasting System (IFS) cycle 48R1
configuration (ECMWF, 2023), using an octahedral reduced Gaussian grid with 159 wavenumbers, or approximately a
horizontal resolution of 69 km, instead of the ECMWF operational configuration which has a resolution of approximately 9
km at present. Also, similarly for affordability  reasons, the experiments only assimilate conventional observations (no
satellite observations), the number of four-dimensional variational (4D-Var) minimizations is reduced from three to two, and
the analysis increments are at a resolution of approximately 210 km (instead of 39 km for ECMWF operations). The
simplified data assimilation setup enables us to run data assimilation experiments for a duration of two months, 01 June - 31
July 1980.

The first experiment is the control. It assimilates the radiosonde observations as vertical profiles. The second experiment
assimilates the radiosonde observations following the balloon trajectory when this information is available (otherwise the
data are assimilated as vertical profiles). The balloon drift in the assimilation is handled by dividing the whole ascent into
15-minute sub-profiles (Ingleby et al., 2018). In each sub-profile, the latitudes, longitudes, and times are invariant. In spite of
this arrangement, which only partially reflects the true slanted nature of the profiles, we retain the terminology of "slanted
profile" when discussing the results, for clarity within this paper.

We consider here the radiosonde observations that were assimilated in both experiments, to ensure no difference in results
may be caused by sampling differences. **Table 4** shows the statistics for these data. For the reasons mentioned earlier, the
interpretation of the table focuses on differences between the two experiments, and not on the absolute level of observation
uncertainties determined by Desroziers' diagnostics. Within 0.1 K, we find no detectable difference between the two
experiments for the levels located below the 100 hPa pressure level. For levels located higher, i.e. pressure lower than 100
hPa, one finds that background departures and estimated observation uncertainties are reduced in the experiment that
assimilated the data along slanted profiles. This result is obtained for radiosondes launched from land stations as well as
radiosondes launched from ships.

The differences may appear as very small and could be discarded as non important, if it was not for the fact that reducing
observation and representation uncertainties is generally an impossible task, once observations were collected and processed
already once. The present findings demonstrate that it is possible to generate greater return, in terms of information content,
through a reprocessing of the observations. The reprocessing enables here to assimilate observations along a slanted
trajectory. Furthermore, these are global statistics - see **Figure 14**. The previous sections indicated that results may vary per
launch site. Consequently, the improvements shown here, for global statistics, must hide some greater improvements at some
particular sites - see **Figure 15**.

Given previous results indicating a larger effect of the balloon drift during winter seasons (e.g. McGrath et al., 2006), and
given the much greater number of radiosonde stations in the Northern Hemisphere as compared to the Southern hemisphere
(e.g., see Figure 12), the present choice of the data assimilation season (Northern hemisphere summer, as Choi et al., 2015)
represents a conservative approach. An impact of larger magnitude may be expected at different time periods, in particular
during Northern hemisphere winter.

## 7. Discussion and conclusions

The verification and evaluation results have shown quite clearly that if at all possible, balloon displacements should be taken into account for all relevant data assimilation applications to minimise representation errors. Ignoring the possibility to account for observation location errors on the 100 km scale would be anachronistic, when global or regional reanalysis data sets approach spatial resolutions finer than 20 km.

The method to reconstruct the balloon position presented in this work is limited by a few assumptions and depends on the vertical resolution of the available profiles, and the conformance of the weather balloons to modern ascent speeds. For the applications tested, an attempt was made to obtain the best results globally, and a clear positive impact was found, particularly when comparing to ERA5 in the early 2000s, although positive results were also found at other times (e.g., 1980s). This is also consistent with other findings in similar settings where trajectory data are used to reduce representation errors (e.g., Laroche and Sarrazin, 2013).

The data assimilation experimental setup employed here is a simplified one, as compared to what may be used in a present-day reanalysis configuration such as ERA5. Yet, we observe a positive impact of the balloon drift in terms of reducing the background departures and the observation uncertainty, using Desroziers' diagnostics, for temperatures in the stratosphere. We expect that the quality of the corrections made to use radiosondes at a displaced horizontal position, as compared to using them at a vertical position, would increase when the background resolution and/or the background quality is increased. In addition, assessing the impact of the balloon drift sensitivity to the assimilation of other observations alongside radiosondes would be worth analysing. However, owing to time and computational constraints, it was not possible to investigate further these effects with full data assimilation experiments at higher horizontal resolution and using all available information, but we note this would be a useful pursuit.

The results of the tests have shown that the method is successful in reconstructing displacements and improving the accuracy of the atmospheric data. Whilst the additional information provided by the method may not always be a visible improvement for individual comparisons, it is of significant value when the displacement changes the gridbox of the model being compared. This has been demonstrated by improved means in the plots and better agreement between observations and ERA5.

The value of improving radiosonde observations by reprocessing of the positions was evaluated by conducting reduced-resolution data assimilation experiments, covering a two-month period in summer 1980. In the future, it would be desirable that the impact of similar activities that seek to improve the observational record be more regularly evaluated in the generation of downstream climate products. Such an evaluation should consider a longer time period and include the impact

on low-frequency variability in the products. For products such as reanalyses, obtained via data assimilation, this should
entail full-resolution Observing System Experiments (OSEs). For other types of climate products, including those powered
by new opportunities such as Artificial Intelligence or Machine Learning (e.g., Singh et al., 2022), it is important that
mechanisms be found to evaluate the impact of using the observations and how changes made in their handling affects the
outcome.

Further experimentation using observation data from the period 2000 - 2020 is crucial and is likely to produce more
compelling outcomes. The effective use of this method for informing future climate reanalysis  is one of the main objectives.
As the world faces increasing challenges related to climate change, the importance of accurate atmospheric data and the
potential of new methods to improve it cannot be overstated. The use of improved position metadata with radiosonde
observations can account for previously unexplainable phenomena, demonstrating the potential of this method to shed new
light on atmospheric data analysis. In addition, the method has the potential to improve the accuracy of reanalyses and
climate predictions, which are crucial for many socio-economic sectors.

To achieve the optimal representation of the data, precise details regarding time and location must be available for every
observation. One significant issue concerns the TAC format's transmission and storage of data, which often only includes a
nominal timestamp such as 00:00 UTC or 12:00 UTC. However, the actual launch of the respective balloon in most cases
took place 30-60 minutes earlier. The precise time difference from the nominal time is frequently unknown, therefore
displacement information cannot be utilised to its fullest extent. Since temperature can vary by more than 1 K/hour in the
boundary layer just due to the diurnal cycle this issue should be addressed. There are well known examples where changes in
the sampling of the diurnal cycle introduced spurious trends into climate data products (Mears and Wentz, 2005). Whenever
possible, the precise launch time should be used. In cases where this information is not available for individual ascents, the
time difference between the nominal and actual launch can often be determined from earlier or later ascents. Operators are
normally advised to minimise the variation throughout the launch procedure and, therefore, launch balloon sondes at the
same time every day.

Additional work to better understand the causes of variation in balloon ascent speeds (e.g., Zhang et al., 2019) could help
further improve the results. Also, given all the uncertainty sources, it could be possible to generate an ensemble of
trajectories for each ascent. Pendulum motion is an effect that would need to be better understood, as it could be of
importance for example in geographical locations where wind advection leads to small horizontal displacements.

The same approach as presented in this paper can be used to reprocess rocketsondes, dropsondes, ozonesondes, or any other
in-situ sonde advected by the wind, provided the necessary information is available. Taking into account the accurate balloon
position would also be beneficial when comparing radiosonde observations with GNSS radio occultation (RO) observations
(Gilpin et al. 2018). Indeed, while it is established practice to consider the tangent point drift of the RO data (e.g., Poli and
Joiner, 2004), radiosonde data is frequently presumed to move vertically only.

In conclusion, the development and testing of the method for reconstructing displacements based on the wind profile shows
promising results. The results presented in this paper suggest taking balloon displacements into account when producing
meteorological or climatological data based on upper-air in situ balloon-borne observations.

## Appendices

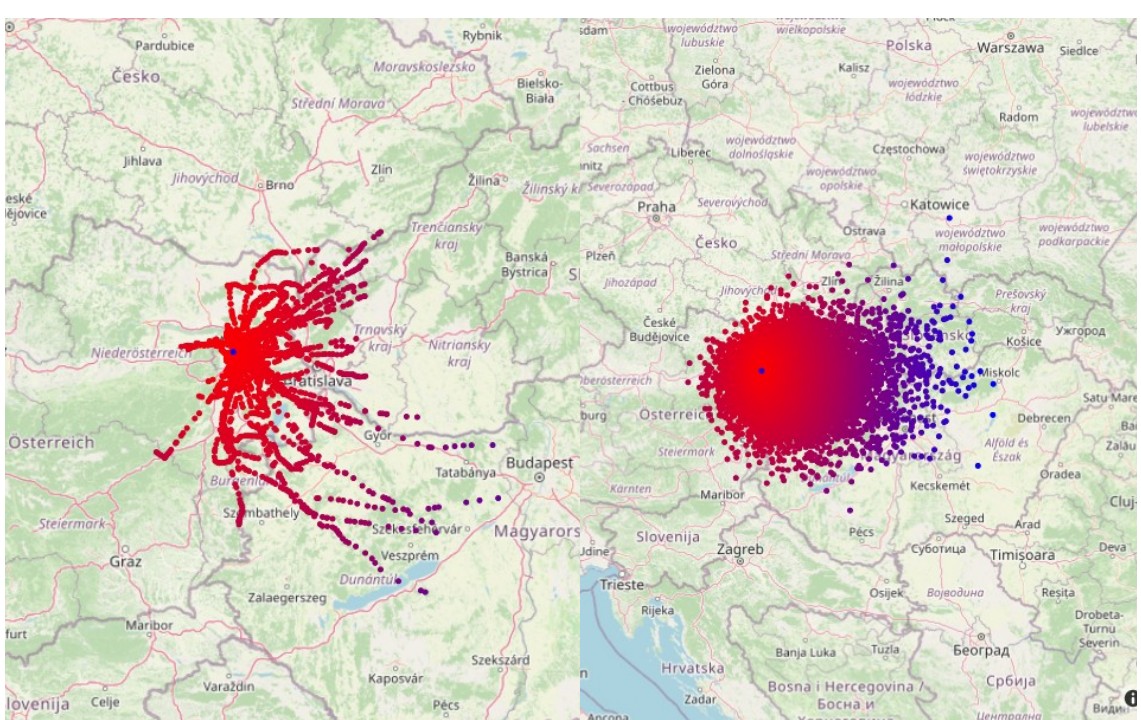

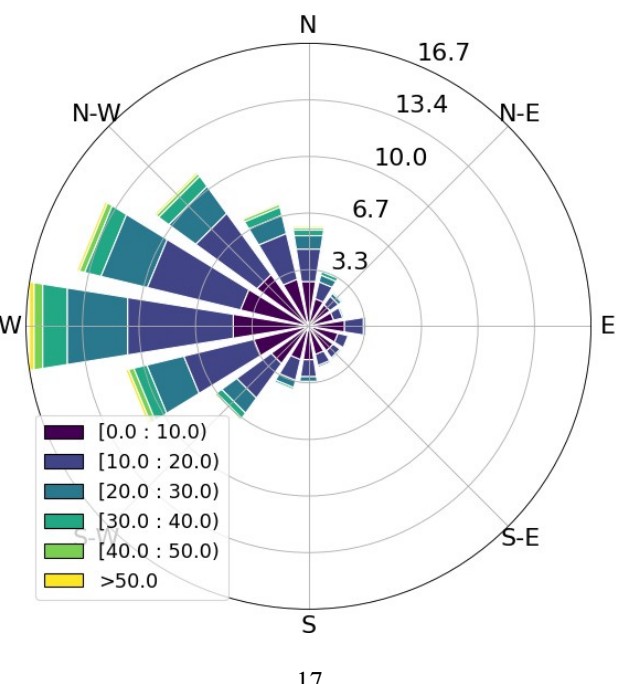

52

**Figure 1: Upper panels: Balloon displacements for station Vienna Hohe Warte, Austria (WIGOS ID 0-20001-0-11035). Central blue dot denotes station location, other dots are balloon positions calculated from wind data as explained in text, coloured red to blue with increasing distance. Note the area covered is non-isotropic around the launch site. Left panel: Trajectories of all radiosonde ascents during the year 2000. Right panel – maximum displacements of all available ascents for all years between 1950 and 2021. Lower panel: windrose of Vienna Hohe Warte station for all available wind data. Colour indicates wind speed [m/s], radius indicates frequency distribution [%] of direction, from where the wind comes from (sectors) and wind speed (colors).**

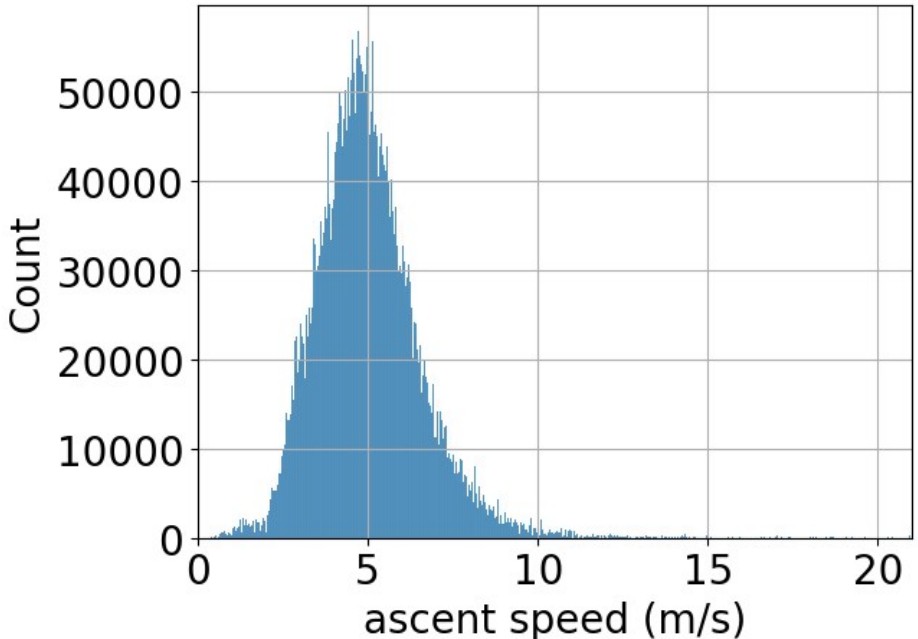

**Figure 2: The observed ascent speeds from a sample of approximately 10 million BUFR encoded observations with known altitude time series in 2020.**

55

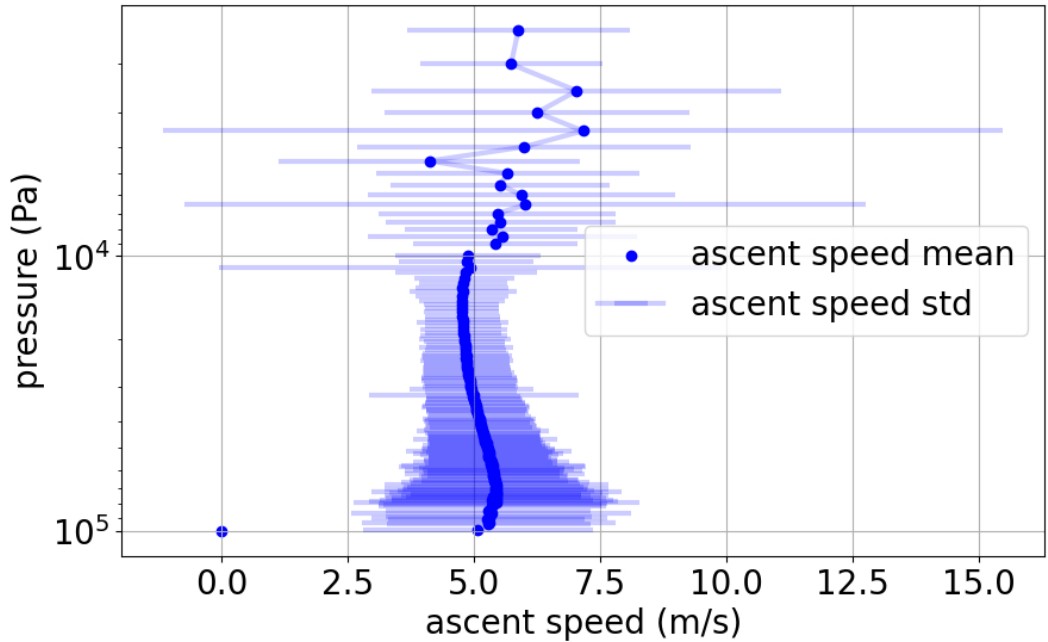

**Figure 3: Mean ascent speed with standard deviation bars for all radiosonde ascents from Riverton USA, in 2020, derived from**
**high resolution BUFR data.**

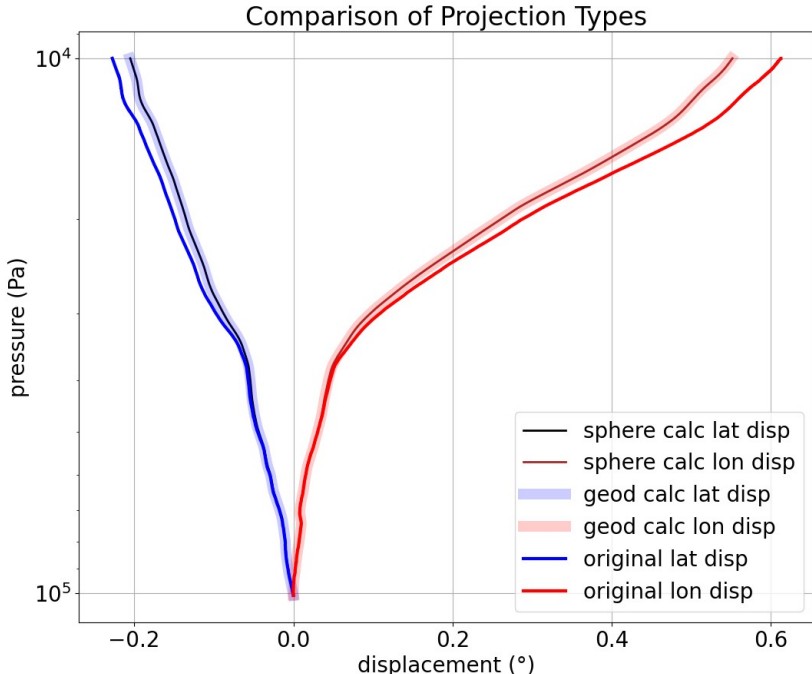

**Figure 4: Calculated displacements (black and brown for spherical earth, thick light blue and red for WGS84). Observed**
**displacements stored in BUFR displacements (blue and red) are included for comparison. Tallahassee, Florida - USA 2020.05.31**
**23:19:00**

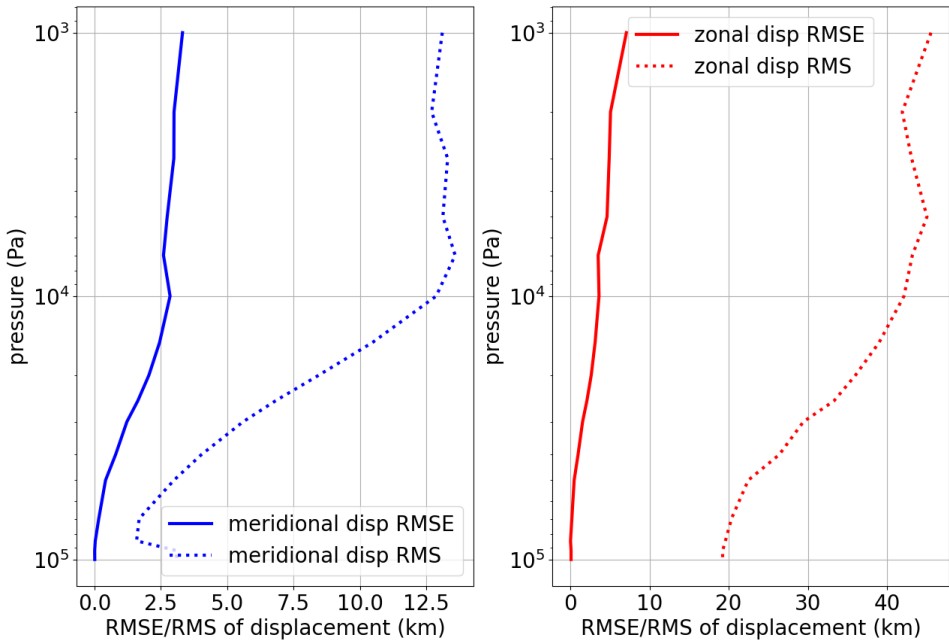

**Figure 5: RMS of meridional (blue dotted) and zonal (red dotted) displacements and RMSE between observed (from GPS) and**
**modelled displacements (solid blue and solid red, respectively). The samples contain all BUFR encoded ascents in the summer**
**months of 2020 (more than 10000).**

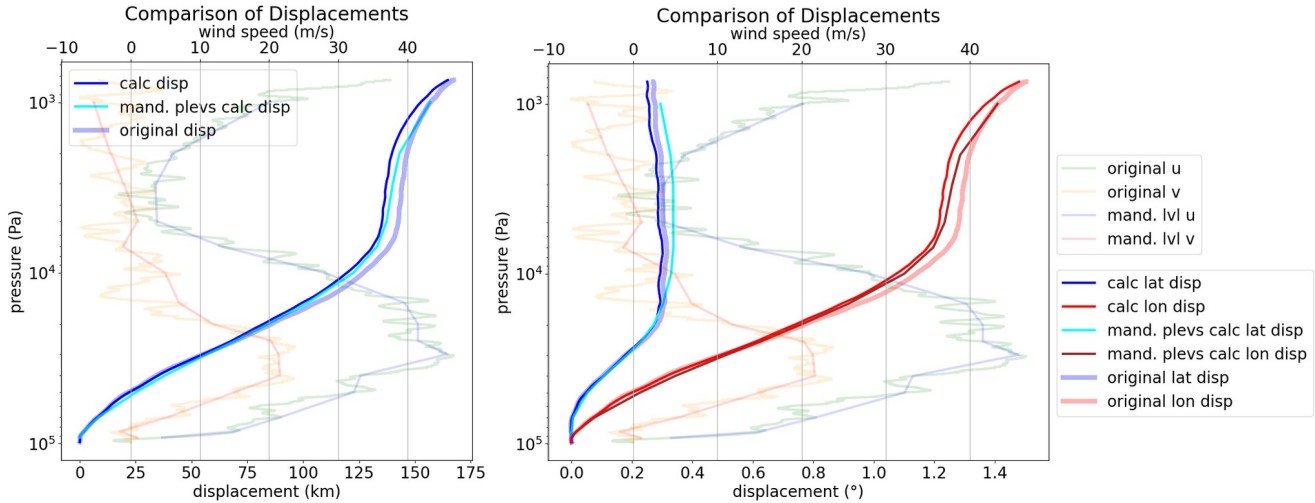

**Figure 6: Vertical profiles of displacements (starting at zero at surface), calculated from observed winds (thin lines) or taken from**
**BUFR thick light lines. The profiles of observed wind (thin light colors) are plotted to the upper x axis - Peachtree City, Georgia -**
**USA 31.01.2021 23:24:00. Left panel: overall displacements in km, right panel: lat and lon displacements in degrees as encoded in**
**BUFR.**

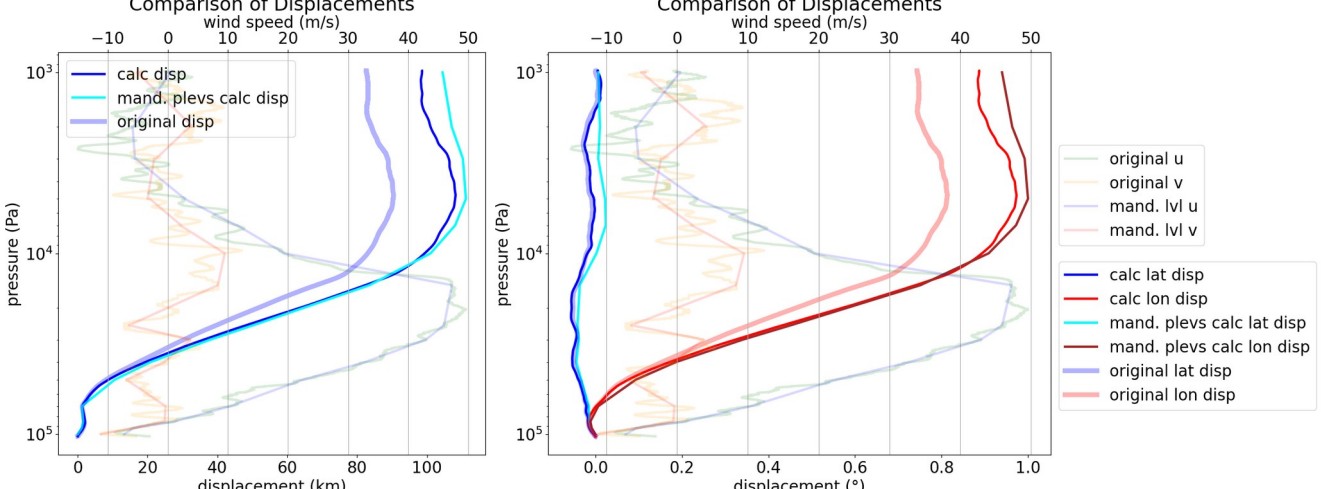

Figure 7: Vertical profiles of displacements (starting at zero at surface), calculated from observed winds (thin lines) or taken from BUFR thick light lines. The profiles of observed wind (thin light colors) are plotted to the upper x axis - Ishigaki, Okinawa - Japan 2019.12.31 23:31:00 . Left panel: overall displacements in km, right panel: lat and lon displacements in degrees as encoded in BUFR.

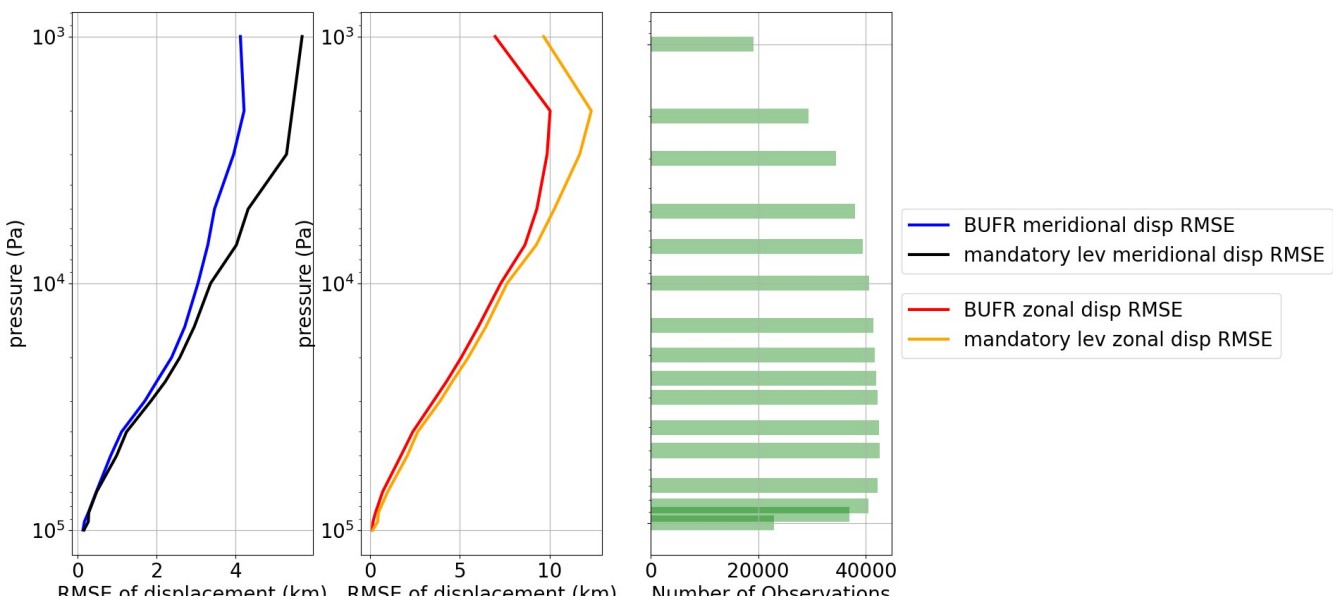

Figure 8: RMSE between observed and modelled displacements of meridional (left panel) and zonal (right panel) components, averaged over all stations available in October 2014, one of the first months with a sizable number of high-resolution BUFR encoded profiles. Blue and red are RMSE profiles obtained by using the full vertical resolution of BUFR observations, black and orange are RMSE profiles, and obtained by using only mandatory level information.

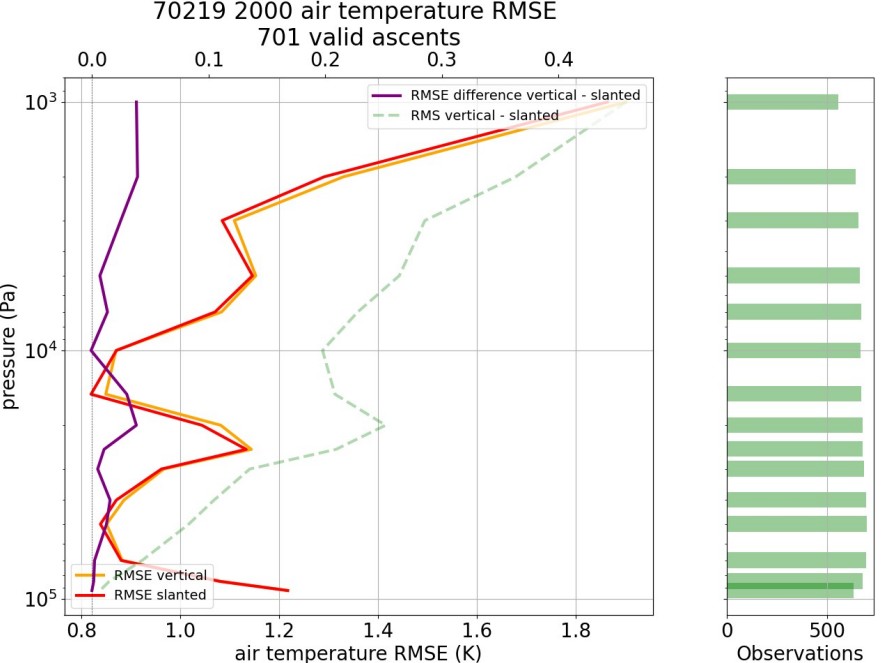

**Figure 9: Bethel Airport, Alaska all 2020 ascents. RMSE (obs - ERA5) of base coordinate temperatures minus sonde temperatures**
**(orange) and RMSE (obs - ERA5) of displaced temperatures minus sonde temperatures (red), also RMS of displaced minus base**
**(green dashed) to show the magnitude of difference between base and displaced temperatures. Positive difference between orange**
**and red graphs (purple line, upper x axis) shows improvement due to more accurate balloon position. Green bars on the right**
**indicate sample sizes at different levels.**

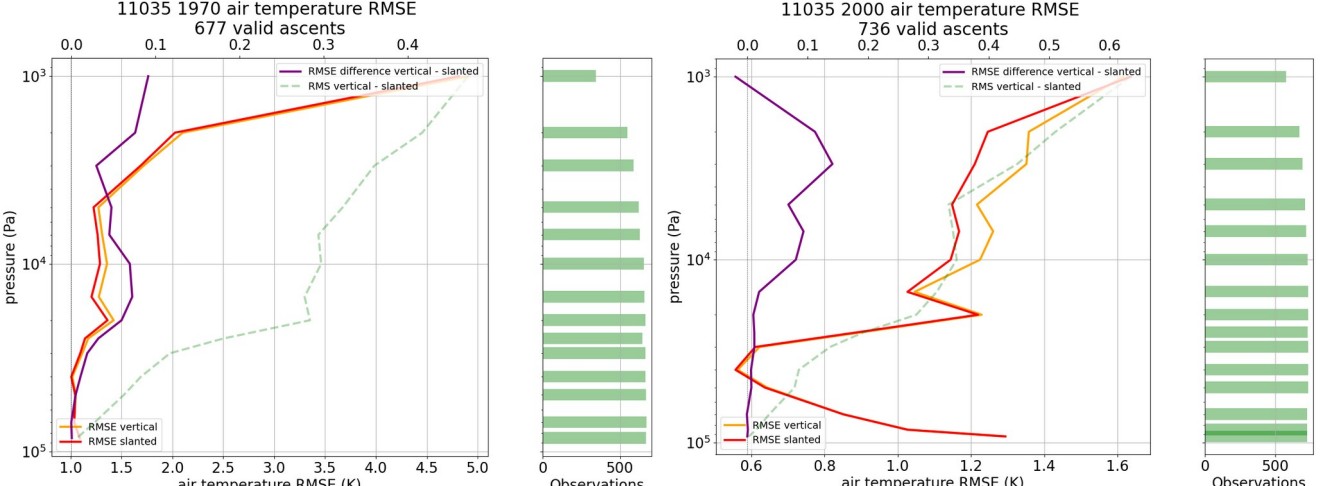

**Figure 10: Vienna Hohe Warte, Austria - Left: 1970 all ascents, Right: 2020 all ascents. Different x-axes scales are used. RMSE**
**(obs - ERA5) of temperature assuming vertical ascents (orange, lower x-axis) and RMSE (obs - ERA5) of temperature from**
**slanted ascents, taking balloon drift into account (red, lower x-axis). Positive difference between orange and red graphs (purple**
**line, upper x axis) shows improvement due to more accurate balloon position.**

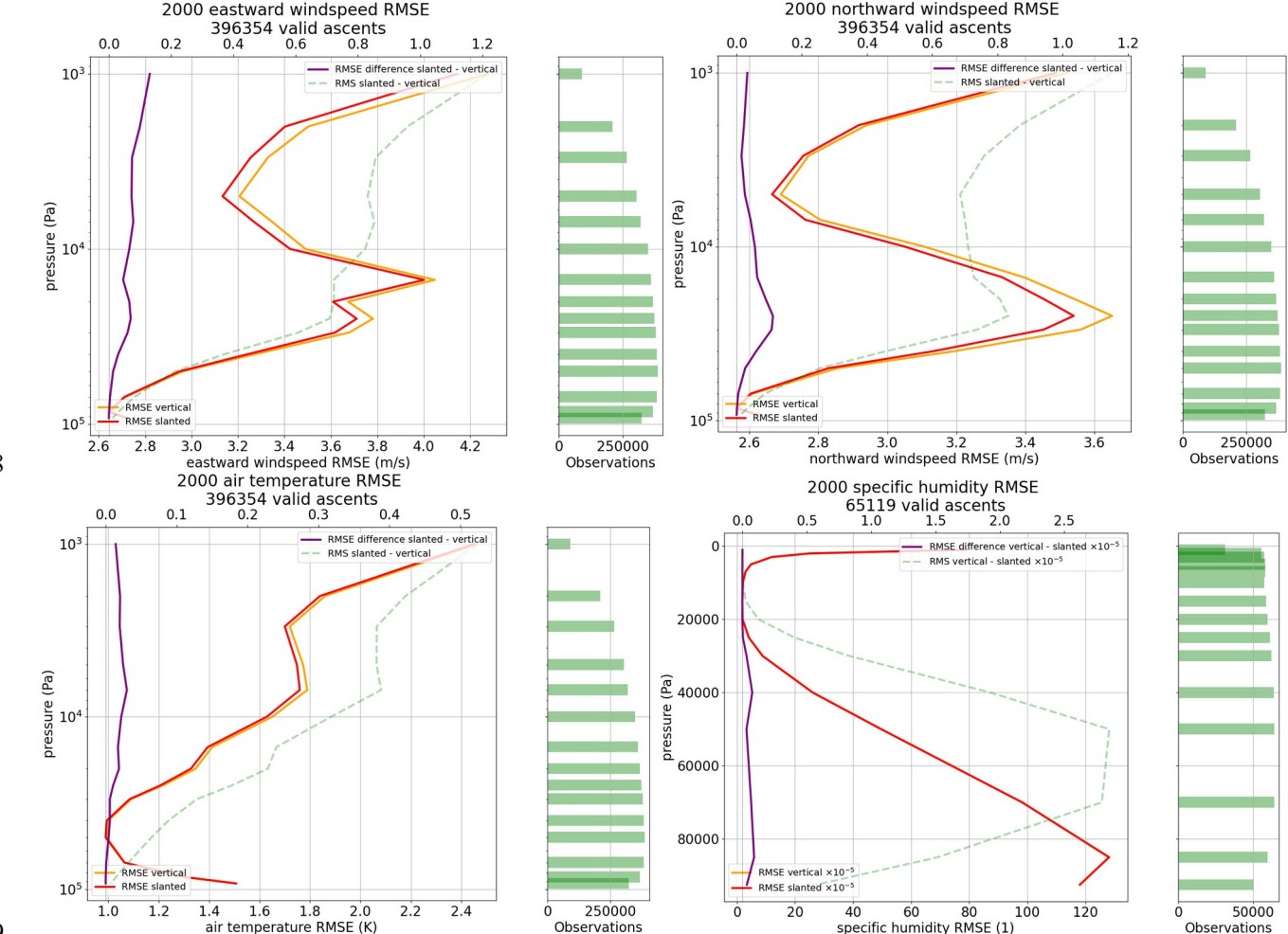


**Figure 11: Global RMSE (obs - ERA5 background) assuming vertical ascents (orange) and RMSE (obs - ERA5 background) from**
**reconstructed slanted ascents (red), calculated from all available ascents of year 2000. The differences between orange and red**
**graphs (purple line, upper x axis) shows how much the better balloon position improved the temperature data (positive =**
**improvement). The "RMS vertical - slanted" (green dashed line, upper x axis) indicates how much the ERA5 background varies**
**on average between the vertical and slanted balloon profiles. - Top left: u wind component; Top right: v wind component; Bottom**
**left: temperature; Bottom right: specific humidity in kg/kg (note scaling factor 10^-5).**

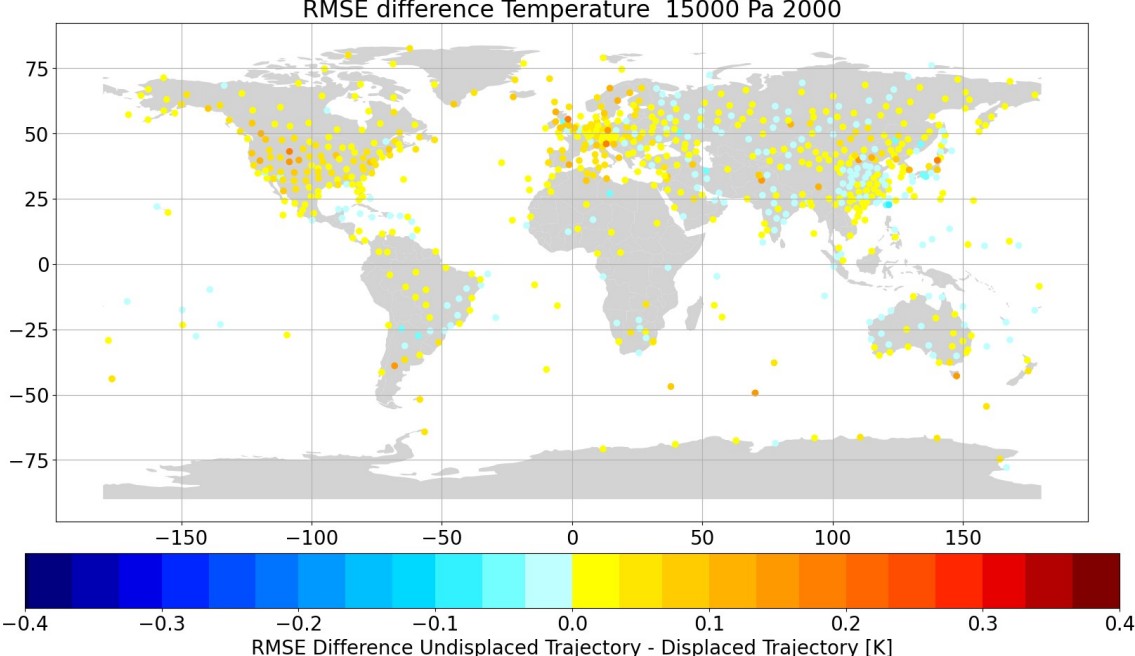

**Figure 12: Global stations difference of temperature [K] observation RMSE (obs - ERA5) when compared to background at**
**station coordinates minus the temperature observation RMSE (obs - ERA5) when compared to background at displaced position -**
**Positive values indicate improvement due to more accurate balloon position. All available observations at 150 hPa averaged over**
**all ascents in the year 2000.**

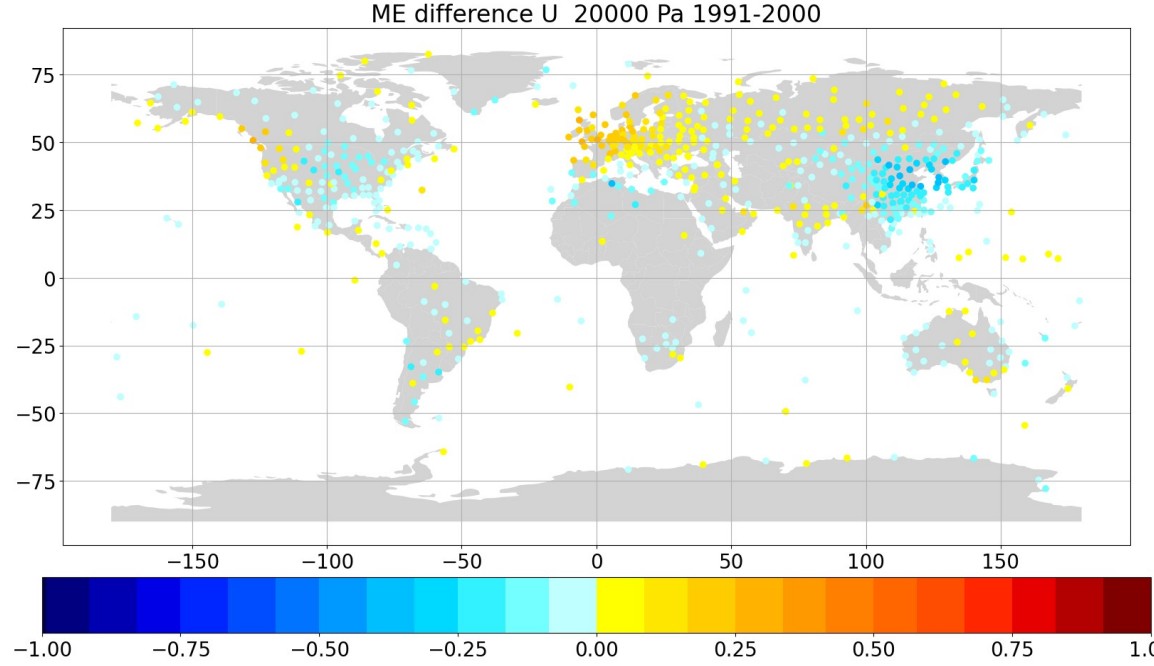

563
**Figure 13: Mean zonal (u) wind [m/s] difference obs - ERA5 background at station position minus obs - ERA5 background at**
**displaced position. All available values on 200 hPa of years 1991 - 2000.**

73

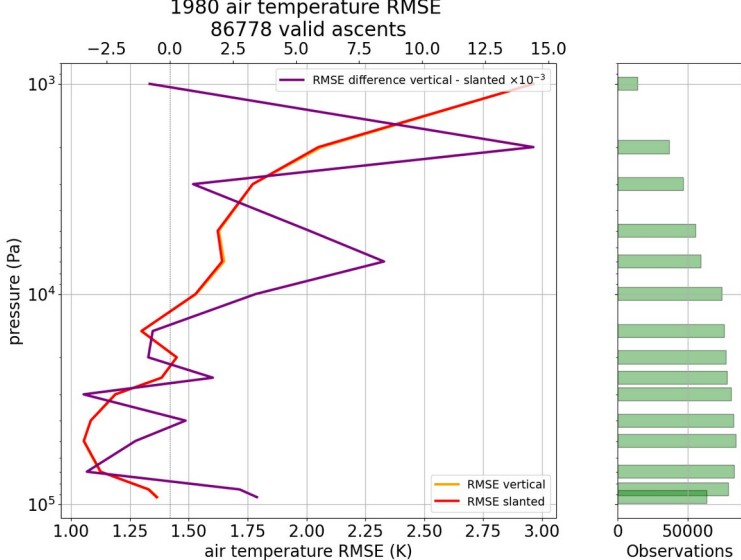

566

**Figure 14: Air temperature obs-bg RMSE difference for experiment "vertical" (orange) and for experiment "slanted" (red). The difference of differences (orange-red) yields the purple line, upper x axis, note scaling factor 10^-3). Positive values indicate improvement due to more accurate balloon position. All available stations on mandatory pressure levels between 1980.06.01-1980.07.31.**

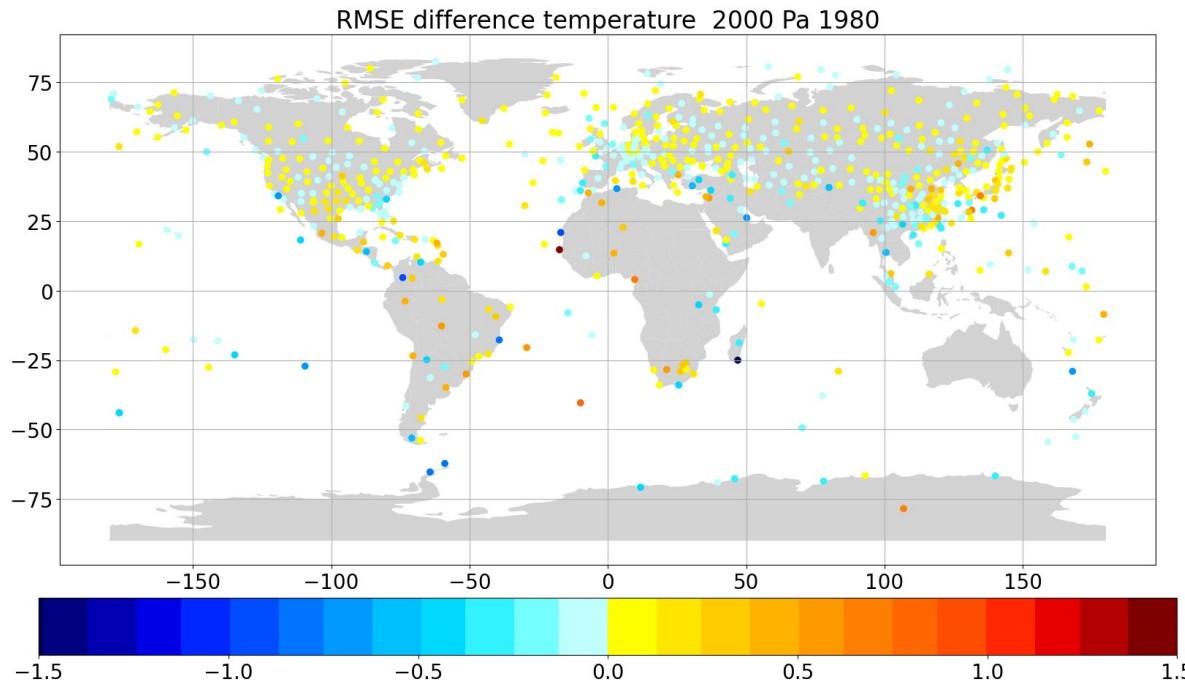

571

**Figure 15: Air temperature obs-bg RMSE [K] difference of experiment "vertical" minus RMSE of experiment "slanted". Positive values indicate improvement due to usage of more accurate balloon position. All available stations on 20 hPa between 1980.06.01-1980.07.31.**

74
75

**Table 1: Ascent speed percentiles for a sample of 10.000.000 observations with known altitude time series in 2020.**

| Percentile | Value | Unit |
|---|---|---|
| 1 | 2.05 | [m/s] |
| 5 | 2.82 | [m/s] |
| 25 | 4.01 | [m/s] |
| 75 | 5.85 | [m/s] |
| 95 | 7.74 | [m/s] |
| 99 | 10.09 | [m/s] |

**Formula 1, 2: Calculation of the vertical gradient of temperature. See Table 2.**

$$\Gamma_{(p)} = \frac{\delta T}{\delta z} = \frac{\delta T}{\delta p}\frac{\delta p}{\delta z} = \frac{-\delta T}{\delta p^{\kappa}}\frac{\delta p^{\kappa}}{\delta p}\frac{\delta p}{\delta z} \tag{1}$$

$$\Gamma_{(p)} = \frac{-\delta T}{\delta p^{\kappa}}\frac{p^{\kappa}}{T}\frac{\kappa g}{R_d} \tag{2}$$

**Formula 3: Calculation of layer height. See Table 2.**

$$\Delta z_{(i \to i+1)} = \frac{T_i}{\Gamma_i}\left(\frac{p_{i+1}}{p_i}\right)^{\frac{-\Gamma_i R_d}{g}-1} \tag{3}$$

**Table 2: Height profile calculation. Explanation of all used variables.**

| Symbol | Description | Unit | Data source |
|---|---|---|---|
| $\Gamma$ | temperature lapse rate | [K/m] | observed variable |
| p | pressure | [Pa] | observed variable |
| T | temperature | [K] | observed variable |
| $\Delta z$ | layer height | [m] | calculated variable |
| $\kappa$ | isentropic expansion factor | [1] | $\kappa = R/cp$ |
| $c_p$ | specific heat capacity of air at constant pressure | [J/kg/K] | constant (1005.7) |
| $R_d$ | gas constant for dry air | [J/kg/K] | constant (286.7) |
| g | standard gravity | [m/s²] | constant (9.80665) |



**Formula 4: Transport of the balloon with the wind. See Table 3.**

$$\vec{s}_{(i+1)} = \frac{\vec{u}_{(i \to i+1)} * \Delta z_{(i \to i+1)}}{w_{balloon}} \tag{4}$$



**Table 3: Time interval calculation. Explanation of all used variables.**

| Symbol | Description | Unit | Data source |
|---|---|---|---|
| $\vec{s}$ | distance travelled | [m] | 0 at i = 0, lon for u, lat for v |
| $\vec{u}$ | wind | [m/s] | observed variable, u and v components of wind |
| $\Delta z$ | layer height | [m] | calculated variable |
| w | rate of ascension | [m/s] | 5, prescribed variable |



**Table 4: Statistics for the radiosonde observations actively used by both data assimilation experiments (vertical and slanted),**
**separating between radiosondes launched from land stations and radiosondes launched from ships. P indicates the pressure (hPa),**
**RSD indicates the robust standard deviation of background departures (i.e., before assimilation), SIGO indicates the estimated**
**observation uncertainty (see text for details), and N indicates the data count. Results that differ between the two experiments are**
**shown in bold and underlined. Observations that were used by only either one of the two experiments are excluded from these**
**statistics.**

| Pressure level range | $P \geq 500$ hPa | | $500$ hPa $> P \geq 100$ hPa | | $100$ hPa $> P \geq 1$ hPa | |
|---|---|---|---|---|---|---|
| **Experiment** | **Vertical** | **Slanted** | **Vertical** | **Slanted** | **Vertical** | **Slanted** |
| **Radiosondes from land stations** | | | | | | |
| **RSD** | 1.2 K | 1.2 K | 1.3 K | 1.3 K | **2.1 K** | **2.0 K** |
| **SIGO** | 1.1 K | 1.1 K | 1.2 K | 1.2 K | **2.1 K** | **2.0 K** |
| **N** | 31,027,909 | 31,027,909 | 30,229,363 | 30,229,363 | 1,358,298 | 1,358,298 |
| **Radiosondes from ships** | | | | | | |
| **RSD** | 1.2 K | 1.2 K | 1.2 K | 1.2 K | **1.6 K** | **1.5 K** |
| **SIGO** | 1.1 K | 1.1 K | 1.2 K | 1.2 K | **1.8 K** | **1.6 K** |
| **N** | 838,265 | 838,265 | 669,655 | 669,655 | 34,709 | 34,709 |


**Code and data availability**

Radiosonde data used in the present work are available from https://doi.org/10.7289/V5X63K0Q (IGRA) and https://doi.org/10.24381/cds.f101d0bf (C3S CDS) and the National Centers for Environmental Information (NOAA NCEI) Radiosonde Archive (https://www.ncei.noaa.gov/data/ecmwf-global-upper-air-bufr/archive/). Climate reanalysis data (ERA5) are available from https://doi.org/10.24381/cds.bd0915c6. The code discussed in this paper is available from https://doi.org/10.5281/zenodo.10663306.

**Author contribution**

Ulrich Voggenberger and Leopold Haimberger designed the method to estimate balloon positions. Ulrich Voggenberger developed the code and optimised the estimations and calculations with further input from Federico Ambrogi. Ulrich Leopold Haimberger and Ulrich Voggenberger validated and evaluated the results based on ERA5 data. Paul Poli ran the data assimilation experiments and evaluated the results in section 6. Ulrich Voggenberger prepared the manuscript with contributions from all co-authors.

**Competing interests**

The contact author has declared that none of the authors has any competing interests.

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
