# Peer review of "Balloon drift estimation and improved position estimates for"

_Geoscientific Model Development, 2023_

## Author Response (AR1)

Line numbers are references to file with edits in suggestion mode.

**RC1:**

General: I am pleased to see efforts to calculate and treat historical radiosonde drift in reanalyses. This is a step is the right direction but needs tidying up.

Thank you very much for the encouraging words!

It is known that radiosondes generally drift further in winter than summer (eg Seidel et al, 2011). This is mentioned in the abstract but nowhere else and I find it a little strange that most of the examples presented are for the summer eg Figure 5 and the assimilation experiments.

This is now mentioned also in chapter 6.
Figure 6 and 7 are replaced by winter data.

Line: 423 + 535

Quality control is an essential part of any data assimilation system (and is still messier than we would like). What quality control (if any) was applied in this study to remove 'bad' reported data?

Quality control is performed at a basic level by threshold checking the input temperature and wind data. The temperature should be between 172 and 372 K, and wind components should be below 150 m/s.

It could be argued that narrowing down the checks based on ERA5 background departure threshold could further clean the data. However, this would make the calculated displacements not independent anymore. The main issue would be erroneous wind values since temperature values are only used for layer height calculation.

We investigated whether further quality control will improve ERA5 RMSE differences:
We removed the 1st and 99th percentile of obs-bg from the ERA5 feedback table, once for each level and once for the whole available wind speed data - both versions did not improve the RMSE differences. Rather it seems that large but realistic displacements are often associated with large departures from the ERA5 bg. In those cases we want to keep the data.

Line: 120

Complication at the poles: The issue is stated in the Provisional 2023 edition of WMO No 8 (see below) and mainly affects the South pole station (which still reports in TAC). If displacements are calculated for this station then the wind direction should also be adjusted.

https://community.wmo.int/en/activity-areas/imop/wmo-no.8/wmo-no-8-provisional-2023-edition
Chapter 13 Measurement of upper wind

Particular care is needed in reporting wind direction near to the North or South Pole especially with the reporting of position at each level. A radiosonde crossing the North Pole must be in a southerly airflow just before crossing the Pole and in a northerly airflow just afterwards. BUFR reports should give wind direction relative to the longitude at each level. (The older alphanumeric codes used a special coordinate system for stations within 1° latitude of the pole.)

You are right it is important to exercise caution when using trajectories from sondes that are likely to cross the poles. This point will also be highlighted in the paper. We will not change the original wind encodings, however,

we only calculate the displacements. If a user finds that the encoding of wind components should change given the new position, he or she may do so.

We also found in WMO CIMO guide part I
https://library.wmo.int/viewer/68695/download?file=8_I_en_2021.pdf&type=pdf&navigator=1 :

"Within 1° latitude of the North or South Pole, surface winds are reported using a direction

where the azimuth ring is aligned with its zero coinciding with the Greenwich 0° meridian.

This different coordinate system should be used by all fixed and mobile upper-air stations

located within 1° latitude of the North or South Pole for wind direction at all levels of the entire

sounding, even if the balloon moves farther away than 1° latitude from the pole. The reporting

code for these measurements should indicate that a different coordinate system is being used in

this upper-air report, in particular if encoded in traditional alphanumeric codes; the location of

the station in BUFR automatically indicates usage of this different coordinate system."

Line: 229

Detailed comments

19 'alongside the estimated wind' - 'alongside the reported wind'?

Yes reported or observed, not estimated - a confusion.

Line: 19

29 'so-called representativeness errors'  There is a recent trend to call them 'representation errors', eg Hodyss and Nichols (2015, Tellus), Janjic et al, (2018, QJRMS) although I would regard these references as optional here.

Edited.

Line: 29

36 'Even later, when GNSS sensors became available, the information collected was often not transmitted' The drift information could not be transmitted in the older alphanumeric codes. It was only with the (ongoing) migration to BUFR that this became possible along with the reporting of many more levels in the vertical (Ingleby et al., 2016)

Ingleby, B., P. Pauley, A. Kats, J. Ator, D. Keyser, A. Doerenbecher, E. Fucile, J. Hasegawa, E. Toyoda, T. Kleinert, W. Qu, J. St James, W. Tennant, and R. Weedon, 2016: Progress toward High-Resolution, Real-Time Radiosonde Reports. Bull. Amer. Meteor. Soc., 97, 2149-2161, https://doi.org/10.1175/BAMS-D-15-00169.1.

Added.

Line: 37

37 'Only since the mid-2010s is the balloon drift taken into account' This needs some rewriting. NCEP have estimated and used radiosonde drift since 2000 (according to Laroche and Sarrazin, 2013), but I haven't seen the documentation for the NCEP method. Laroche and Sarrazin (I think) implemented it in the Canadian NWP

system. Ingleby and Edwards (2014, ASL) at the Met Office were early users of the BUFR drift positions and quoted an earlier study: 'Macpherson (1995) found little benefit from treating balloon drift and concluded that for simple advection without wind shear the errors due to neglecting the exact time and position of each level cancelled.'

Suggested text: 'In the 2010s there was a move towards using the balloon drift information in modern observation processing of GNSS sondes, with beneficial results (e.g. Laroche and Sarrazin, 2013; Ingleby et al., 2018).'

Used the suggested text.

Line:39

60-62 'High-resolution radiosonde data ... in BUFR ... University of Wyoming' For information ECMWF BUFR radiosonde data is available (loosely associated with IGRA) at the following link:
https://www.ncei.noaa.gov/data/ecmwf-global-upper-air-bufr/archive/
There is a separate archive (slightly less comprehensive) of data received by NCEP.

Thanks for the suggestion, while we extracted the BUFR data from the MARS archive, they are identical to those from the NCEI source. We included the link to the data source list, since they can be downloaded from there by everyone.

Line: 67, 645,

73 [GNSS sondes] 'can track the horizontal and vertical position of the balloon and the sensor at high frequency' Delete 'the balloon and'. The high frequency data, especially the raw data (before smoothing), sometimes clearly shows the effect of pedulum motion under the balloon (eg Ingleby et al, 2022 https://doi.org/10.5194/amt-15-165-2022 and references, the results there raise questions about how optimal the smoothing is) - worse because of the long suspensions often used now (55 m is recommended for the RS41). I feel that pendulum motion should be briefly mentioned.

In our experimental cases, we did not observe pendulum motion, at least not on the scale of the displacements considered. However, we added a note on this important point in the revision and included this reference.

Line: 91

76 'if a single theodolite was used' I am sure that the vast majority of theodolite ascents only use a single theodolite.

Early observations used only ground-based tracking, e.g., by theodolite, which was fairly accurate but could lose the balloon early during cloudy or strong wind conditions, and relied on an assumed ascent rate if, **like in most cases,** a single theodolite was used (e.g., Favà et al., 2021).

Line: 78

89 '[BUFR] has a much higher vertical resolution (1 second frequency' This is potential resolution. Many European ascents report with 2 second frequency, there area also many reports from other areas (eg China) with fewer than 300 levels.

Clarified that.

Line: 104

NB. Perhaps slightly clearer to replace 'the latter' with 'BUFR'.

Done.

Line: 103

91 '5-degrees'

Done.

Line: 106

103 '2.3 Estimation of the balloon trajectory ' There is some overlap with the work of Laroche and Sarrazin (2013). Mention this in the text and explain where there are differences.

Done.

Line: 238

119-120 ', but temporal information is not available for all the reported pressure levels.' - ': time information is not available for any of the reported pressure levels' (slightly clearer)

Done.

Line: 154

130 'a piecewise polytropic atmosphere' I had to look up 'polytropic' and I am still not quite sure what it means in this context. I suspect that you mean that temperature is a piecewise linear function of height - please say this if so. My preference is to avoid the word polytropic.

Yes the temperature gradient is assumed to be constant between the layers.
Will be reformulated.

Line: 167

155-156 'This is true for the differences within a single ascent, but also for the differences between different accents.' - 'This is true both within a single ascent and also between different ascents.'

Done.

Line: 192

192 'The software necessary for the creation of calculated balloon trajectories can be found in: ' Is it the same software in each of the three repositories?

All repositories contain identical code.
We removed the github link, it is findable via both other links.

Line: 247

215 '4. Verification with GNSS radiosondes' 'Validation' might be better than 'Verification'

Changed all occurrences of verification to validation.

Lines: 269 and other occurrences

216 'indisputable reference' I suggest replacing indisputable with 'good'. I agree that GNSS data is usually very good quality, but there can be problems with interference or only a few GPS satellites above the horizon. As mentioned above pendulum motion complicates a detailed view of the positions/raw winds.

Done.

Line: 270

230 'Figure 5 ...'  The longitude displacements (in degrees) are larger than the latitude displacements - presumably the longitude displacements are dominated by high latitude ascents. I recommend showing values in km instead and ideally show summer and winter displacements separately.

Longitudinal displacements are greater than latitudinal displacements, not only in high latitude ascents. The westerlies and trade winds are predominantly zonal.

Regarding the summer month examples, two plots were replaced by winter data. Additional plots of the total displacements (in km) were added for clarification and comparison. Also this plot was re-done in zonal and meridional distance in km. Other plots were kept in ° to preserve consistency with BUFR encoding.

Line: 530

At the lowest level there is a local maximum in latitude RMS - probably caused by a minor error in the reported launch position, I have seen such errors.

This seems to be a very plausible explanation.

Y-axis - personally I would prefer to see more levels labelled.

This should stay unchanged to keep all logarithmic plots in the same layout.

246 'Figure 8' Again, I think that displacements in km would be better.

Done. Now in zonal and meridional displacements in km.

Line: 557

333-334 'The second experiment assimilates the radiosonde observations as slanted profiles' Is each level given a separate latitude/longitude/time or is it done (for efficiency reasons) in 15-minute sub-profiles as in Ingleby et al (2018).

Indeed the assimilation follows the approach implemented and documented by Ingleby et al. (2018). We have clarified this in the text. We have kept the word "slanted" for consistency with the rest of the paper, explaining the particular difference.

Line: 400

334 'the balloon trajectory, when it is available' ie when there is a reasonably complete wind profile?

Yes, correct. When the lowest observation is not too far above ground and the available levels are not too far apart, trajectories will be calculated.
The boundaries are now specified in the revised version.

Line: 130

364 'Laroche, 2013' - 'Laroche and Sarrazin, 2013

Line: 440

370-371 'we also expect that any positive impact of sonde displacement be amplified when satellite observations are used alongside radiosondes' I'm not convinced of this - more satellite data usually means less impact of radiosonde data (especially temperatures).

We agree that increased reliance on satellite observations reduces the influence of the radiosonde data in the analysis. As the background quality improves, thanks to the greater quality of the previous analyses after ingestion of satellite data, then the radiosonde departures with respect to the background are reduced in statistical metrics. There are several aspects to be considered if one wants to extrapolate about the impact depending on the use of other observations. First, the representation errors that are caused by balloon drift, if unaccounted for in the assimilation, would remain identical whether or not the assimilation system used many satellite data. In such a case, representation errors would then explain a relatively larger fraction of the departures, in a system that assimilates many satellite data, as compared to a system that does not. Second, assuming that the analysis step does not resolve representation uncertainties, removing representation errors before presenting the background departures to the assimilation would allow the analysis to fit the radiosonde observations better, thereby contributing to the observations having more impact. Lastly, if the background quality improves, one may also expect the realism of horizontal gradients to improve in the background, and then the corrections in background departures caused by balloon drift handling to be *also* of higher quality, thereby removing a larger fraction of the representation error, as compared to within a data assimilation system that has a poorer quality in terms of background horizontal gradients. All these points are reasons why we postulated that the impact of the balloon drift may be 'amplified' in a system where other sources of errors are reduced thanks to assimilation of satellite data. That said, these are all arguments that ignore the systematic component of the uncertainties, and, also, probably more importantly, the structured components, such as some locations being often exposed to prevailing winds that advect the radiosondes across stable gradients. In the end, so many factors can influence the results, e.g. via feed-back loops in the bias correction of satellite radiances, that these discussion points can remain hand-waving arguments, until proven with due numerical experiments (a number of which were already published by other authors, as referenced). We hence removed such statements in the revised version and remained on a factual level.

Line: 445

384 'experiments, covering a two-month period in summer 1980'  There is usually more drift in winter.

As is often the case with data assimilation experiments, which are computationally expensive undertakings, the exact choice of time period was influenced by considerations of technical nature. In the present case, the availability of a number of pre-existing experiments for summer 1980 with the same version of the IFS as used for the present investigations presented an important advantage. This time period is indeed important for climate reanalysis as summer 1980 is the first season of well-constrained troposphere (and stratosphere, albeit to a lesser extent) thanks to starting availability of two TOVS satellites.This pre-existing material enabled us to run the radiosonde drift data assimilation experiment for longer than would have been the case otherwise, if baseline experiments had been needed prior to establishing a control experiment. We also note that Choi et al. (2015) also ran experiments in the summer season (2013 in their case), albeit for a much shorter time period (under a month). All that said, we agree that, under ideal circumstances, one should run data assimilation experiments over several time periods. Nevertheless, the present choice of season is important, even if it is at the lower end of the expected impact. A negligible impact would indeed indicate that the effects are not worth considering in

data assimilation for this particular season. This is not what we find, and for this reason we believe the results are worth presenting. We have added a statement in the revised paper.

Line: 423

419-420 'reconstructing displacements based on the reversed calculation of wind speed and direction' 'reconstructing displacements based on the wind profile' would be better in my opinion. With GNSS the winds are sometimes derived from the Doppler effect rather than differentiating the displacements.

Line: 504

We now use the suggested text.

Figure 1. Perhaps clarify that the wind rose gives the direction from which the wind is coming.

Done.

Line: 517

Figure 9. At first I didn't understand what the green dashed line was. I now think it is the RMS difference between ERA5 temperatures 'vertical' and 'slanted' - I presume that there are time differences included too (please confirm). Does the dashed green line use the upper or lower axis?

Correct, it is the RMS difference between vertical and slanted. There are no time differences included - the 3-hourly gridded ERA5 data used for this does not have sufficient time resolution to include this aspect. The green line uses the upper axis.

Figures 11. The 10 hPa differences from ERA5 in 1970 are very large, presumably due to the radiosonde type used at the time and possibly a lack of quality control. I wondered if figures 10 and 11 add much or would be better omitted.

We want to keep Figure 11 to demonstrate @370-371, while Figure 10 can be omitted.

Line: 570

Figure 15. The 10 hPa obs-bg RMSE values are huge (15 degrees). My guess is that there has been little or no attempt to quality control the reported values.

The legend now states that the obs-bg RMSE values need to be multiplied with $10^{-3}$, indicating their small size.

573 'Favà et al' drop 'Discuss.' ?

Done.

Line: 684

**RC2:**

I am glad to see this work being done so that rawinsonde data can be better assimilated into re-analyses. One thing that should be clarified is the two different datasets (TAC and BUFR), that one contains time and three-dimensional position information and one does not, and the fact that perhaps not all data are yet transmitted in BUFR. Therefore, this technique could also be useful in operations until the time that all data are available in BUFR.

This has been added.

Line: 38, 106

I note that similar work has been done for downsondes (as opposed to the current upsondes), https://doi.org/10.1175/JTECH-D-17-0023.1, and that this should be referenced.

Done.

Line: 238

Some of the writing is confusing and could use some editing for improvement. Lines 240-242 are a good example.

To enhance readability, we will begin with this explanation and utilise simpler sentences with a logical progression of information.

Line: 293

Some additions to the current study would be helpful:

1. Mean statistics with height over all sondes in both TAC and BUFR of the differences between the calculated and actual trajectories. Figure 5 shows the statistics for the "summer months" (undefined) for one year, which is helpful, but incomplete. In addition, it isn't clear why the displacements are separated into latitudinal and longitudinal components instead of total distance. If there is a physical reason to do so, this should be explained. If there is no physical reason, then total differences should be shown.

We will plot the total displacements additionally for clarification but otherwise use lat/lon as units of displacement to maintain consistency with BUFR encoding. For Figure 5 and 8 we now show zonal and meridional displacement in km now.

Line: 530 and other occurrences.

2. Perhaps examples of particularly accurate and particularly inaccurate displacement calculations (trajectories) can be shown to show the user how this may work in practice.

Figure 6 and 7 have been updated to provide examples, which show the range of accuracy of the calculated trajectories. This is also mentioned in the text.

Line: 535 and 292

Additional comments:

Figure 1 is not referenced in the text.

Done.

Line: 30

Significant levels are defined starting on line 92. There are also significant levels for thermodynamic variables, and these should also be defined here for completeness.

A short indication of their existence has been added.

Line: 111

The text says that PILOT/PIBAL have height data. All observations with mandatory levels have height data (if there are thermodynamic data), but it seems that these values are not used in the calculations. If this is correct, why not? If this is incorrect, clarify in the text.

It is used to calculate observation time for PIBAL ascents. It is not used when temperature is available, since this ensures consistent calculation of height across all radiosondes. In historical ascents the geopotential has been often calculated differently, using different values for gravity acceleration etc. This was one reason why since the 1980s geopotential height has no longer been assimilated in most NWP centres. Therefore we avoid using it as well.

Line 198: Is the station height also necessary for the calculation?

Station height is not necessary for the calculation but it is used for quality control purposes. Profiles that begin more than 1500m above the station height are discarded as it is assumed that too much information is missing.

Station height can be used for the calculation of "time since launch", whenever lower levels are missing. The missing wind information does not allow a displacement calculation, but the estimated time since launch can still be reconstructed.

Line: 132, 161

---

## Author Response (AR2)

Line numbers are references to file with edits in suggestion mode.

**REVIEWER 1**

The current version is improved over the original, but I have a good number of questions about the study that should be addressed:

1. I assume that the data shown in Fig. 1 are from calculations based on this work, but that isn't clear.

We have changed the figure legend such that it is clear that the dots shown in upper panes are calculated from the IGRA data used in this study.

504

2. Line 109: Does this temperature limit vary with height? 373K is possible near the surface, but not near the tropopause, for example.

We agree this is very crude and would remove only extreme outliers mostly by coding errors. These do not occur in practice because such outliers are already set to missing within the IGRA2 quality control system https://www.ncei.noaa.gov/data/integrated-global-radiosonde-archive/doc/igra2-product-description.pdf.

We therefore now write: For temperature, needed for geopotential calculations, we relied on the IGRA2 quality control (Durre et al. 2018) that already removes gross errors. An additional very crude check (temperature between 173 and 373K) was applied just to verify that the data were read correctly.

113

3. Line 192: How is "improvement" measured?

Calculated displacements are verified against GNSS measured displacements, using the difference as error metric. The differences found when using a polynomial fit for ascent speeds were only negligible smaller than when using a constant ascent speed.  We modified the text accordingly.

205

4. Lines 232-234: I believe that the older data are available in TAC format, which is in kt, and in degrees and speed. Is there code to do the conversion from TAC format to what is needed here, or a way to directly input the older data without conversion?

This is correct. Since there must be code to read from different formats, like IGRA or directly from TAC or other archives anyway, the software relies on the users to do these unit conversions.

5. Figure 4: It is interesting that meridional displacement differences are much smaller than zonal ones. Is this because the zonal wind is usually faster than the meridional wind?

Yes, that is definitely the reason, particularly in the jet stream regions. The majority of stations can also be found in the mid-latitudes.

6. Figure 8: It would be useful to know how many ascents are used at each level.

The figure now contains the number of ascents as in the other figures.

541

7. Section 5: I believe that the ERA5 assimilates the rawinsonde data as vertical profiles, which should be mentioned.

"This is to be expected, since radiosondes were assimilated as vertical profiles in ERA5." was present in line 329.

8. Lines 309-310: I think you mean comparing the radiosonde observations as slanted profiles, not the forecasts.

We meant that the better fit to ERA5 can be found only with background forecasts, not the analyses, since those were drawn to the measurements at the wrong place. To make that a little clearer, we added (in contrast to analyses) to the sentence.

9: Line 522: I'm not sure what "vertical temperature" is. Is it the ERA5 temperature?

This was indeed not well formulated. We changed this passage now to "RMSE (obs - ERA5) of temperature assuming vertical ascents (orange, lower x-axis) and RMSE (obs - ERA5) of temperature from slanted ascents, taking balloon drift into account (red, lower x-axis)."

555

10. Figure 13: The caption says u (zonal wind), but the description in the text just talks about speed. Is this total wind speed, u-speed, or zonal velocity?

This has now changed to "zonal (u)" wind, to make it clear that only the u-wind component is meant.

576

Minor and pedantic comments:

1. When upper air is used as an adjective, sometimes it is hyphenated, sometimes not. It should always be hyphenated when it is an adjective (upper-air measurements), but not when it is a noun (which is never the case in this version of the manuscript).

It is now upper-air throughout the text.

2. Distinguish between wind (the motion of air) and quantification of that motion (wind speed, wind velocity, etc.). For example, line 77, the wind velocity is calculated.

Done.

3. Line 114: What is "ERA5 feedback?"

We removed feedback, which is actually just a technical term for some statistics calculated during assimilation. This has now been changed to "ERA5 background forecast differences"

4. Line 121: What are the "input data," and what is it put into?

"input data" is now replaced by radiosonde data for the displacement calculation

5. Base coordinates is defined twice (lines 125 and 133)

Second definition is now removed.

145

6. Table 3 is mentioned before Tables 1 and 2.

Changed.

7. Line 182: The ascents are the same, whether the data are reported in high-resolution or not. Suggest changing to something like high-resolution reports or high-resolution data (note that high-resolution should be hyphenated when it is an adjective.)

We changed to high-resolution data

8. Line 536: Figure 12 is labeled Figure 13.

This is now corrected.

576

**REVIEWER 2**

General: Quality control.

I am pleased to see that the results in section 6 use the standard ECMWF QC. It slightly reduces the independence of the background, but it does 'clean up' the data. One point that I forgot to mention originally is that non-GNSS Russian radiosondes can have large height errors at larger displacements and low radar-elevation angles (this would affect the comparison of all the variables with model fields) this is a consequence of relying on radar heights without a pressure sensor.

See Kats A., Balagourov A & Grinchenko V. (2005) The impact of new RF95 radiosonde, introduction on upper-air data quality in the North-west region of Russia. Poster Pw(07), TECO-2005, WMO/TD- No. 1265; IOM Report- No. 82.

Thanks for this comment. We added the following text to the paragraph on quality control:

"The reason was not always the displacements themselves but also the fact that large lateral displacements can lead to large height errors in profiles from non-GNSS Russian radiosondes, since those have no pressure sensor but rely on radar heights (Kats et al. , 2005). However, even for these sondes, we found that taking into account the balloon drift reduces the differences to the ERA5 background forecasts."

The two figures below show temperature RMSE profiles at two radiosonde stations using Russian sensors (22543 used a MRZ-3AK1 with MARL-A radar, 36003 used a PAZA-22M with MARL-A radar). The left panels show the differences for ascents with maximum displacements less than 0.5 degrees, the right panels for ascents with maximum displacements above 0.5 degrees. For the larger displacements we get better agreement by taking into account the drift. However this does not remove the height error problem, which appears to cause the peak in rms errors around 300 hPa, which can be seen at many Russian radiosondes.

[Figure]

Winds near the poles

The topic is mentioned now, which is a step in the right direction, but I think the issue of adjusting the wind direction should be discussed directly. My suggestion: "The WMO Manual on Codes states that for stations within 1° of either pole wind direction shall be reported in such a way that the azimuth ring shall be aligned with its zero coinciding with the Greenwich 0° meridian. There is currently an attempt to update this advice for BUFR reports, such that wind direction should be reported relative to the current reported longitude - to help in NWP use of such winds. Before comparing winds from the South Pole station with NWP fields they should have their direction adjusted when the drift positions are calculated (not currently done)."

The comparison of 'South Pole' winds with the ECMWF background winds could be looked at before and after drift calculation for the experiment in section 6. I am fairly sure that no NWP system uses that 1° convention when outputting model winds near the pole so _someone_ needs to adjust the wind direction.

It occurs to me that there are possible issues at other stations too, eg 71082 launches at 82.5°N, a 220 km drift could be about 15° longitude - should a direction adjustment be made?

Thanks for this comment. We implemented your suggested text. You are right that also for the northernmost stations this effect may lead to noticeable wind direction errors and should also be taken into account, but we did not mention that in the text.

Detailed comments

31. 'steep horizontal gradients' - 'sharp horizontal gradients' slightly better

Done

32

40. '(Ingleby, 2018)' - '(Ingleby et al., 2018)'

Done

42

68. 'procedural errors' Optional: could also mention the work of Gandin and Collins eg. Collins, WG, 2001: The operational complex quality control of radiosonde heights and temperatures at the National Centers for Environmental Prediction. Part II: Examples of error diagnosis and correction from operational use.

JOURNAL OF APPLIED METEOROLOGY 40, pp 152-168

Done, but we did not this reference, since it is a bit redundant to the literature already cited.

82. 'in the used input databases' - delete 'used'

Done

85

154. 'between the layers of the profile' - 'between the levels in the profile'

Done

168